# NEUTRAL RESIDUES:
# REVISITING ADAPTERS FOR MODEL EXTENSION

## ABSTRACT

We address the problem of extending a pretrained large language model to a new domain that was not seen at training time, like adding a language for which the original model has seen no or little training data. Popular solutions like fine-tuning or low-rank adaptation are successful at domain adaptation, but formally they do not add any extra capacity and degrade the performance in the original domain.

Our paper analyzes this extension problem under three angles: data, architecture and training procedure, which are advantageously considered jointly. In particular, we improve adapters and make it possible to learn an entire new language while ensuring that the output of the neural network is almost unchanged in the original domain. For this purpose, we modify the new residual blocks in a way that leads each new residual block to output near-zeros in the original domain.

This solution of *neutral residues*, which borrows architectural components from mixture of experts, is effective: with only 20% extra learnable weights compared to an original model trained on English, we get results that are significantly better than concurrent approaches (fine-tuning, low-rank or vanilla adapters) in terms of the trade-off between learning a new language and not forgetting English.

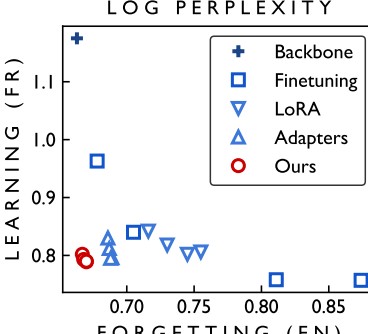

Figure 1: **Learning or Forgetting?** Fine-tuning a model reduces the model performance on the original task. Potential trade-offs are governed by the learning rate, with extreme cases being (*top-left*) the original backbone and (*bottom-right*) finetuning on the new data with a large learning rate.

LoRA or vanilla adapters mitigate catastrophic forgetting but only to some extent. Our method significantly improves this compromise. Detailed results in Section 4.

## 1 INTRODUCTION

The dominating strategy for producing foundation models involves training from scratch on a large collection of data, typically trillions of tokens, covering multiple domains. Training such models is extremely costly in terms of computing resources. In view of the explosion of the number of gigantic models produced in the last years (the HuggingFace model repository claims to host about one million models), this training paradigm raises the problem of the economical and ecological sustainability of the model production pipelines. As an example, training the Llama 3 model is estimated by multiple sources to cost hundreds of millions dollars, mostly due to the training on 24000 flagship GPU*s* H100 for months[1], not counting the hidden costs of exploration.

In this paper, we consider the question of extending an existing model, in order to add some new capabilities or knowledge without retraining it from scratch and therefore in a much more resource-

---

[1] https://www.theinformation.com/articles/ten-gifts-nvidia-gave-its-investors

efficient manner. The solutions that are successful for domain adaptation, such as finetuning or Low-Rank Adaptation (Hu et al., 2022, LoRA), are not adequate in this context because they do not add any extra capacity. Therefore they are inherently limited in the amount of knowledge that they can incorporate without suffering significant forgetting. This catastrophic forgetting is a well-known problem in the continual learning setting (McCloskey & Cohen, 1989; French, 1999).

In contrast, the adapters introduced by Rebuffi et al. (2017) extend a pretrained model by adding new parameters. When transferring to a new domain or task, only these new weights are trained while those of the original backbone are frozen. This extension strategy, which was initially introduced in computer vision with convolutional neural networks, was subsequently adapted to textual transformers by Houlsby et al. (2019). Adapters benefit from the initial pretrained model and hence exhibit competitive transfer performance, typically on par with finetuning. They enjoy several additional properties: Only a subset of weights is trained. Most importantly, they formally add some capacity and therefore do not suffer from the aforementioned limitations. However, adapters still suffer significant forgetting in their current form and are therefore not sufficient for our problem.

In our paper, we build upon adapters and, accordingly, increase the capacity of the model by adding feed-forward blocks to the model. In particular, we consider the use-case where we add a new language to a pretrained model. We measure how the extended model performs both on the training criterion, namely perplexity, but also on downstream tasks such as question answering.

We then analyze how to optimize the compromise between learning a new language, and not forgetting the initial knowledge of the pretrained model. We study the impact of (1) training data; (2) training strategy, such as designing loss specifically intended at reducing the forgetting, and (3) architecture. This leads us to identify several important factors for successfully extending a model with adapters:

- **Data:** When learning adapters, a simple way to drastically reduce the forgetting is to keep training with a small fraction of data similar to the original distribution.
- **Architecture:** An *adapter gating* mechanism is an effective way to ensure that the network distinguishes when it should operate as the original neural network or when the new blocks are desirable to process the input data.
- **Initialization:** Our analysis concurs with the observation by Houlsby et al. (2019) that near-identity initialization is important when training adapters. We introduce a variant that is even more drastic that the existing 0-block initialization of the output matrix.
- **Training:** The gating mechanism is advantageously informed in a supervised manner. We proposed two strategies in that respect, both introducing a local loss: The first is inspired by Mixture-of-Experts and involves an explicit domain classifier at each block. The second (and most effective) one involves a sparsity loss whose objective is to ensure that the residual connections output near-zero values when the input follows the pretraining distribution.

Overall, we show that multiple ingredients are advantageously intertwined to ensure that the extended model still approximately implements the original function when the input is from the original distribution, without being too constrained to incorporate a large amount of new knowledge. In our experiments, these conclusions are validated with three models by extending the pretrained model with two languages.

## 2 RELATED WORK

The success of deep learning is often associated to the outstanding performance obtained when training convolutional neural networks on large datasets such as Imagenet (Deng et al., 2009). Yet another key benefit is their amenability to transfer knowledge between tasks with a limited amount of training data (Oquab et al., 2014). This transfer is usually performed with a so-called *finetuning* stage, where the original backbone is trained in a data-efficient manner to solve a distinct target task than the one employed at training time. It often requires to modify the architecture to adapt the output layers to the new tasks. Several self-supervised pretraining methods are noticeably interested in improving ways of pretraining models on proxy tasks (Devlin et al., 2018; Caron et al., 2021), such that networks finetuned from these backbones, or used in a 0-shot manner, perform well on multiple tasks. In most of these settings, the backbone only requires some form of domain adaptation.

Finetuning is however not sufficient when a large body of knowledge must be incorporated to the network. In this section we review a body of literature related to this problem and to our work. This includes solutions that add new knowledge into a pretrained network, as well as methods that incorporate gating mechanisms such as mixtures of experts (MoE).

**Parameter Efficient Finetuning.** Enhancing language model capabilities using module based training has gained interest in recent years due to the high cost of full finetuning. Those methods are known as parameter efficient finetuning methods (PEFT; Houlsby et al. (2019); Hu et al. (2022); Li & Liang (2021)). Unlike full finetuning, they only require a limited amount of memory. Houlsby et al. (2019) proposed to insert small bottleneck layers, known as Adapters, within each transformer layer, allowing models to be finetuned by training only a small fraction of the parameters while keeping the original model frozen. Low-Rank Adaptation (LoRA) by Hu et al. (2022) adds trainable low-rank matrices to transformer layers while keeping the original weights frozen.

**Continual Learning without Forgetting.** Continual Learning is a widely studied concept (De Lange et al., 2021; Wang et al., 2024) as it allows the addition of new knowledge to LLM after their initial pretraining. It is usually done through full finetuning in order to learn domain specific knowledge (Roziere et al., 2023), to enhance instruction-following abilities (Wang et al., 2023) or to align LLM with human preferences (Ouyang et al., 2022). One of the biggest challenges of continual learning and the main drawback of doing it through full finetuning is catastrophic forgetting (Kirkpatrick et al., 2017). This is partially alleviated by using the initial training data (Robins, 1995), when available, or as a proxy data from a similar distribution.

Several studies have investigated more advanced methods to address forgetting in continual learning (Biderman et al. (2024); Wu et al. (2024); Li et al. (2024); Zhong et al. (2024); Riemer et al. (2019); Li & Hoiem (2017)). Biderman et al. (2024) show that using LoRA helps reducing forgetting but is less efficient in learning than full finetuning especially for distributions far from the LLM pretraining distribution. Wu et al. (2024) proposed a continual learning method for LLMs, which amounts to adding new intermediate transformer blocks in an interleaved manner. This enables the injection of new knowledge while preserving the initial capabilities.

**Mixture-of-Experts.** MoEs have a long history in neural networks (Yuksel et al., 2012), for instance with set of classifiers where each individual classifier was specialized in a different task or aspects of data. Residual architectures (He et al., 2016) and in particular transformers (Vaswani et al., 2017) have renewed the interest in this strategy. In this context, they amount to replacing the standard feed-forward networks (FFN) of the transformer blocks by a collection of FFNs, referred to as experts. A gating mechanism selectively activates a subset of relevant experts for a given input. This technique has been widely used to reduce the computational cost of inference in large language models (Jiang et al., 2024; Xue et al., 2024) with a focus on the design of an expert-balancing loss to promote equitable token distribution across experts (Shazeer et al., 2017; Pfeiffer et al., 2023).

Recent works have explored utilizing MoE's gating mechanism to mitigate catastrophic forgetting during the continual learning phase of language models. LorRAMoE (Dou et al., 2024) employs LoRAs as experts, integrating them with a gating mechanism to improve performance on specific downstream tasks through Supervised finetuning (SFT). LoRAMoE mitigates catastrophic forgetting in the case of SFT, but to our knowledge its effectiveness has not been studied in a broader context. In contrast, two recent parallel concurrent works are closer to our proposal: Zhong et al. (2024) augment Mixture-of-Experts LLMs with new modalities by addition of new experts to a pretrained mixture of experts. Similarly, Li et al. (2024) add an MoE layer in parallel to the FFN of the transformer block to learn multilingual capabilities and mitigate catastrophic forgetting during Continual Learning. Our solution shares some similarities with this work by employing mixed data training. Yet Li et al. (2024) do not integrate an explicit sparsity criterion, and need to weight differently the blocks of the pretrained model with those of the gated adapters. They also consider multiple elements in the mixture and apply a softmax for the rooting mechanism. Since at least one expert is activated in their mixture, this selection mechanism also prevents the capacity of the model to produce 0-valued outputs. Our solution of neutral residues with local loss is more effective than simply gating adapters, as shown in our ablation Section 4.4.

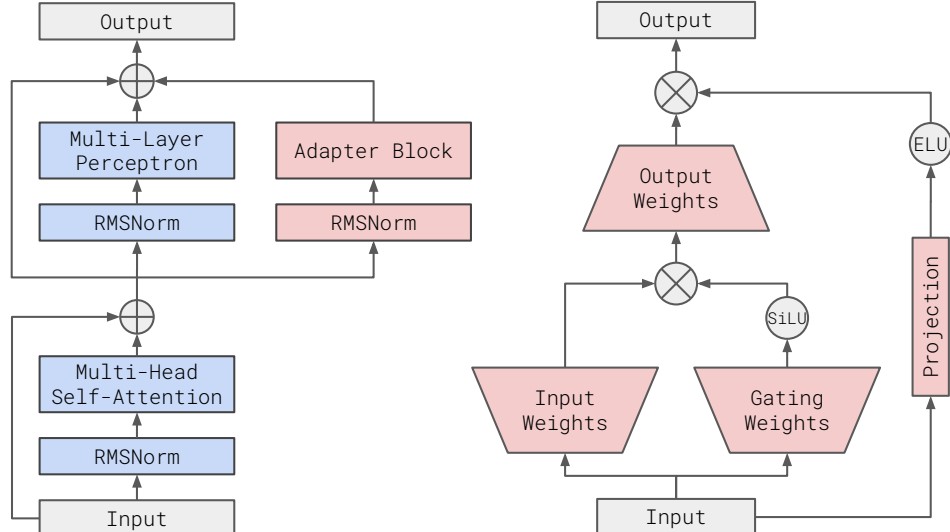

Figure 2: Architectural design adopted for neutral residues: (*left*) We add the adapter block in parallel to the FFN. (*right*) Our gated adapter block include a gate with a ELU linearity (see Section 4.4 for a comparison with a sigmoidal gating). At training time, the trainable parameters are represented by red blocks. The weights of the blue blocks are frozen. The output of the adapter is optimized with a local loss such that the output is sparse if the input follows the pretraining distribution.

# 3 NEUTRAL RESIDUES WITH GATED ADAPTERS

The initial adapters proposal by Rebuffi et al. (2017) was initially introduced in the context of convolutional residual networks. Their motivation was to adapt a backbone to multiple tasks by finetuning on a limited amount of data, rather than adding a large body of knowledge. Therefore they consider a limited number of trainable parameters, typically less than 10% additional weights compared to the original backbone. In the context of language models, the serial adapter from Houlsby et al. (2019) adds residual feed-forward block sequentially after each multi-head attention and the original feedforward. In our case and as shown in Figure 2, we use parallel adapters as we did not observe a significant difference with serial adapters. This concurs with observations by Touvron et al. (2022) that blocks can be parallelized pairwise if vision transformers are large enough.

**Near-to-identity initialization.** One key ingredient pointed out by Houlsby et al. (2019) is that it is best to start training adapters such that the initial network is near-identical to the pretrained one, thereby not modifying the initial function. This is typically done by initializing the output matrix to $0s$, as advocated in particular by Zhang et al. (2019). We are even more drastic in that respect: in addition to setting the output matrix with $0s$, we depart from the usual He's initialization for both the input and gating matrix and initialize them with a much lower variance, such that the model remains longer close to the original one. More specifically, we reduce the variance employed in He's initialization: instead of using variance $2/d$, where $d$ in the input dimensionality of the matrix, we use a variance of $1/(d \cdot L)$, where $L$ is the number of transformer layers.

**FFN vs multi-head attention (MHA).** In our experiments carried out for adding a new language, it is more effective to add extra weights in the form of FFN than in MHA blocks, for a given number of additional parameters. This concurs with how extra capacity is added in MoE. Similarly Li et al. (2024) present an ablation who show that adding FFN blocks is significantly better than adding attention blocks. Note, this is in contrast with what is observed by Hu et al. (2022) with the LoRA method, for which the best choice to adapt to different downstream tasks with few parameters is to update the MHA matrix, in particular they chose to update the keys and values matrices. In our preliminary experiments, we re-evaluate LoRA in our context of knowledge addition, and observe that LoRA is significantly more effective when updating the FFN blocks than the MHA blocks. This apparent discrepancy is likely explained by the fact that domain/task adaptation is different from the question of adding a large body of knowledge.

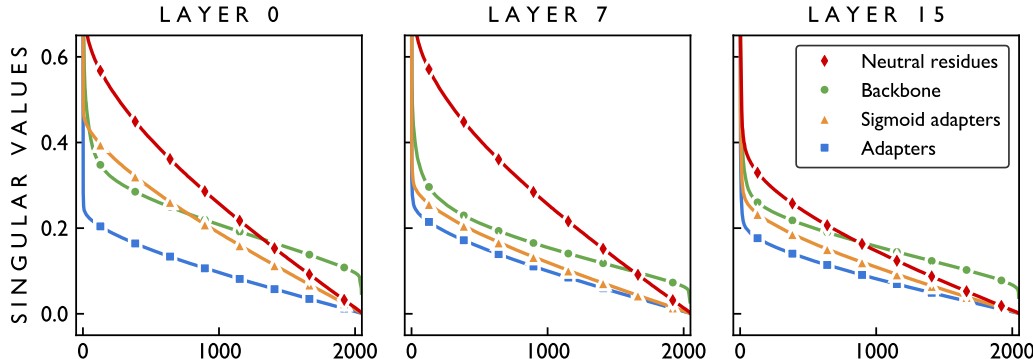

Figure 3: **Spectral analysis** of the gating matrix, normalized by the largest singular value.

**Mixed distribution training.** As discussed in the related work, one solution to limit catastrophic forgetting is to keep training with a proportion of the original data, if such data is available. This strategy is typically employed with finetuning but is also effective with all baselines. Li et al. (2024) report that catastrophic forgetting almost disappear when the volume of original language data is five times greater than the new one, yet they also point out that mixed data training is likely to slow down the learning of the new language.

In our work, we assume that we have access to a "similar distribution" at posttraining time when we learn the adapters. We utilize a proportion $p$ of data aimed at being similar to the original distribution (but not identical), which in our case is data in English. We also consider a case in which we do not know perfectly what kind of data was used at training time, like in the case of the Gemma et al. (2024) model. In our ablation section 4.4, we evaluate the impact of $p$ on learning and forgetting. As a particular case, we also analyze the case $p = 0$, where no pretraining data is available or used.

**Adapter gating & local loss.** Recent transformer models commonly adopt a gated activation with a SiLU non-linearity (Elfwing et al., 2018). The gating weights implement a linear operator $W_g$. When training a vanilla adapter block for a new language, the singular values of this operator are significantly more skewed than in regular transformer block from the pretrained backbone. Figure 3 shows this behavior, which is detrimental to the learning process. Our interpretation is that all projections instantiated by this operator tend to be colinear and therefore to behave similarly when combined with the SiLU non-linearity, because they implicitly operate as a old/new data switch. This hinders the capacity of the model.

To address the problem, we add a block gate as in MoE transformers (Shazeer et al., 2017) and two recent works on model extension with gated MoE (Zhong et al., 2024; Li et al., 2024). In our case, we use a single gate whose role is to operate as a selector, either explicit or implicit, coupled with a local loss applied at the block level. This gating is combined with two distinct strategies:

- *Explicit selector with sigmoidal activation.* This solution is inspired by MoE. In this case, the block gate is associated with a sigmoid activation. It is therefore trained as a local binary classifier whose objective is to distinguish between the old and the new data distributions.

- *Implicit selector: ELU activation with sparsity constraint.* This strategy, which is the one that we refer to as "neutral residues" unless specified otherwise, does not make an explicit classification. Instead, we let the ELU activation adapts itself the strength of its response for the block. We also apply a loss $\ell_1$ on the output of the adapter governed by a hyperparameter $\alpha$, which favors the adapter to return a residues of $0s$. It is only employed when feeding the extended model with data of the original distribution.

As one can observe in Figure 3, the gating operator has a significant effect on the singular values of the gating matrix: the distribution of the singular values is less skewed than the original backbone. The implicit selector is more effective than the local classifier in that respect. As we will see in Section 4, it is the method that offers the best compromise between not forgetting and learning.

# 4 EXPERIMENTS

In this section, we report experimental results to validate the performance of neutral residues to adapt a neural network to a new domain. We consider the case of adding or improving the multilingual capacity of a large language model. More precisely, we start from an English-only model (or a model that has only seen a small amount of non-English data) and finetune it on French data.

## 4.1 DATASETS AND EVALUATION PROTOCOLS

**Training Datasets.** For the French finetuning dataset, we use a dataset extracted from Common-Crawl, which was pre-processed with the following steps. First, the text content was extracted from HTML using the `resiliparse` package. Then, we performed language identification with `fastText` to keep only French data. Next, we performed deduplication at the paragraph level, by computing a hash of each paragraph. Finally, we filtered document using a `fastText` linear classifier, which was trained to discriminate random pages from CommonCrawl *vs.* high-quality documents such as Wikipedia articles, textbooks, news articles or scientific articles.

For the English domain, we decided to restrict ourselves to text from Wikipedia. The motivation for this is to use a finetuning dataset that is close, but yet different, from the training dataset of the backbone model. Indeed, as discussed earlier, in many cases one does not have access to the original dataset that was used to train the backbone model, and we want to demonstrate that our method is robust even in the case where exact data from the original domain is not available.

**Evaluation benchmarks.** We consider two main ways to evaluate the performance of finetuning. First, we evaluate the perplexity of the model on held-out sets, both in English and in French, to measure forgetting and learning. For English, we consider text from a domain that does not correspond to the finetuning data, and use the PubMed subset from ThePile (Gao et al., 2020). The goal here is to evaluate how robust our method is to forgetting, especially in the case where we *do not have access* to the same training distribution as the original model. For French, we consider text from the same distribution as the finetuning dataset.

Second, we consider standard academic benchmarks used to evaluate large language models, such as question answering or Cloze-style problems. We use the following datasets: ARC challenge (Clark et al., 2018), HellaSwag (Zellers et al., 2019), MMLU (Hendrycks et al., 2020), CommonSense QA (Talmor et al., 2018) and Belebele (Bandarkar et al., 2023). For French, we use the translated datasets from Dac Lai et al. (2023) and Sakai et al. (2024). For all datasets, we score each answer independently as the continuation of the question, and predict the most probable one. We use normalization, following the recommendation from Gu et al. (2024). We perform 5-shot evaluation for ARC challenge, CSQA and MMLU, and 0-shot evaluation for HellaSwag and Belebele.

**Backbone models.** We consider three models that we use as backbone. The two first models were trained internally on 2.4T tokens, and have 1B parameters. The main difference between these two models is that the first model, denoted **Transf-EN** was trained on English data only. The second model **Trans-ML** is MultiLingual: it was trained on a mix of English, French, German, Spanish, Italian and Portuguese data. The non-English data represents roughly 12.5% of the tokens. The last model is **Gemma-2B** (Gemma et al., 2024). It also includes a small amount of multi-lingual data during pre-training. We do not have access to the training dataset of Gemma, making it a realistic test case for our approach. Appendix A gives more details about these models.

For finetuning with new knowledge, we train during 100000 steps with a batch size of 64 for Trans-ML and Trans-EN, and we train Gemma-2B during 150000 steps with a batch size of 8.

**Baseline extension strategies.** In our experiments, we consider the following methods:

- *Finetuning:* we train all the weights from the backbone. We re-initialize the optimizer state.
- *LoRA:* unlike Hu et al. (2022), we add all the additional parameters in the FFN layers, as in our preliminary experiments this choice was significantly better for knowledge extension.
- *Adapters:* we use the vanilla adapters as Houlsby et al. (2019) with near-identical initialization. We use parallel instead of serial blocks to have a more directly comparable baseline.

Table 1: **Mixture of distributions:** Impact of the rate $p$ of data that follow the pretraining distribution when training a vanilla adapter. 10% of initial data distribution is a good compromise between learning (FR ppl) and not forgetting (EN ppl).

| English rate $p$ | EN | FR |
|---|---|---|
| 0.00 | 0.720 | 0.810 |
| 0.01 | 0.707 | 0.810 |
| 0.10 | 0.687 | 0.812 |
| 0.50 | 0.683 | 0.828 |
| Pretrained model | 0.663 | 1.175 |

Table 2: **Training with new data only or mixed data.** Starting from an English pretrained model, we measure the tradeoff in between learning and not forgetting if we only post-train with French or if we keep $p = 10\%$ of English. We consider full finetuning, LoRA and vanilla adapters baseline, and our neutral residues. See Table 1 for results of the pretrained model. We report the perplexity on the validation sets at the end of training.

| Method | $p = 0$ | | $p = 0.1$ | |
|---|---|---|---|---|
| | EN | FR | EN | FR |
| Finetuning | 0.874 | 0.755 | 0.811 | 0.758 |
| LoRA | 0.770 | 0.814 | 0.730 | 0.818 |
| Vanilla adapters | 0.720 | 0.810 | 0.687 | 0.812 |
| Neutral residues | 0.684 | 0.790 | 0.668 | 0.793 |

**Hyperparameters.** Except mentioned otherwise, LoRA and both adapters use 20% of extra learnable weights. We use a learning rate of $5 \cdot 10^{-5}$ by default, except for finetuning since the tradeoff between learning and not forgetting is better with a learning rate of $2 \cdot 10^{-4}$, see ablations in Section 4.4. The hyperparameters $\alpha$ and $p$ have been selected based on perplexity results on the validation set. We provide other training hyperparameters in Section A.

## 4.2 Preliminary Analysis

**Training with mixed data distribution** Table 1 analyzes the impact of the proportion $p$ of data from the pretraining distribution used to extend the model, measured for vanilla adapters. As one can see, $p = 0.1$ offers a good trade-off: the forgetting is not significantly higher than with $p = 0.5$, while the model is almost as good in French as when training only with French ($p = 0$). Therefore, when training with mixed data, we set $p = 0.1$ in all subsequent experiments.

Table 2 gives the trade-off between learning and not forgetting for all baselines and our method, when training either without or with mixed training distributions. As one can see, the mixed training significantly reduces the forgetting for all the methods without hindering too much the learning.

Table 3: **Impact of the number of parameters.** We vary the extra capacity allocated to new learnable weights in our method. As to be expected, the capacity of our adapters to perform well in the new language strongly depends on the added weights. In contrast, the impact of extra weights on forgetting is almost neutral.

| Extra weights | Perplexity | | Tasks avg. | |
|---|---|---|---|---|
| | EN | FR | EN | FR |
| +5% | 0.667 | 0.849 | 46.8 | 39.5 |
| +10% | 0.667 | 0.819 | 46.2 | 40.8 |
| +20% | 0.668 | 0.793 | 46.5 | 41.2 |
| +50% | 0.669 | 0.761 | 46.7 | 42.3 |

## 4.3 Main results

Table 4 reports our main detailed results when we want to improve the performance in French of a pretrained model. Table 5 provides the counterpart when the new language to be added to the pretrained model is German, which overall leads to the same conclusions. As one can see, our proposal is overall offering the best trade-off between learning and not forgetting compared to all other techniques: it is first in terms of the sum of both metrics (English+French or English+German).

**Impact of the initial model.** First, one can observe that the results are consistent for all models. This indicates that even if we don't know exactly the data employed to produce the pretrained model, our neutral residues is beneficial. Note, this is also evidenced by Table 2, where neutral residues perform significantly better than other techniques even when we do not use any pretraining data. If we compare Transf-EN with Trans-ML, as expected the pretrained model Trans-ML obtains much better performance on French (resp. German). However, even the Trans-ML model, which has seen some multilingual at training time, significantly benefits from our posttraining method.

Table 4: **Main results:** comparison of neutral residues (*ours*) versus four baselines for different models. All methods except finetuning use 20% of trainable parameters compared with the backbone models. We use a proportion $p = 0.1$ of French for the training. We report the average performance across all tasks both on English and French. These metrics reflect the overall performance with respect to not forgetting and learning, respectively. In **bold** we report the two best models.

| Model | Method | Forgetting: English | | | | | Learning: French | | | | | Task avg. | |
|---|---|---|---|---|---|---|---|---|---|---|---|---|---|
| | | BeleBele (0) | HellaS (0) | Arc C (5) | CSQA (5) | MMLU (5) | BeleBele (0) | HellaS (0) | Arc C (5) | CSQA (5) | MMLU (5) | English | French |
| Transf-EN | Backbone | 43.0 | 60.3 | 40.5 | 56.5 | 34.5 | 34.6 | 32.8 | 25.9 | 37.4 | 27.4 | **47.0** | 31.6 |
| | finetuning | 37.9 | 45.5 | 34.8 | 45.5 | 31.0 | 42.2 | 48.6 | 34.5 | 57.0 | 31.7 | 39.0 | **42.8** |
| | LoRA | 42.0 | 53.1 | 38.1 | 52.3 | 33.2 | 42.4 | 46.2 | 33.0 | 51.5 | 30.7 | 43.7 | 40.8 |
| | Adapters | 42.1 | 56.9 | 38.7 | 53.6 | 33.4 | 43.2 | 46.2 | 31.7 | 50.0 | 30.9 | 45.0 | 40.4 |
| | *ours* | 43.4 | 59.5 | 39.9 | 55.4 | 34.3 | 43.6 | 48.5 | 32.2 | 50.4 | 31.4 | **46.5** | **41.2** |
| Transf-ML | Backbone | 41.2 | 59.4 | 40.6 | 57.4 | 34.4 | 42.6 | 48.9 | 34.4 | 47.6 | 31.1 | **46.6** | 40.9 |
| | finetuning | 40.1 | 52.3 | 37.2 | 54.3 | 32.0 | 43.8 | 52.7 | 40.4 | 57.8 | 32.6 | 43.2 | **45.5** |
| | LoRA | 40.0 | 57.1 | 39.8 | 54.1 | 33.5 | 44.2 | 51.1 | 37.4 | 53.1 | 31.7 | 44.9 | 43.5 |
| | Adapters | 41.2 | 58.0 | 39.0 | 55.1 | 34.6 | 41.8 | 50.0 | 38.0 | 55.4 | 31.4 | 45.6 | 43.3 |
| | *ours* | 42.1 | 59.3 | 40.6 | 55.3 | 34.2 | 42.9 | 51.1 | 38.1 | 53.7 | 32.1 | **46.3** | **43.6** |
| Gemma-2B | Backbone | 46.3 | 69.7 | 47.0 | 63.0 | 39.9 | 44.3 | 50.3 | 36.8 | 43.8 | 32.5 | **53.2** | 41.5 |
| | finetuning | 43.7 | 55.2 | 38.8 | 52.6 | 34.3 | 45.2 | 55.3 | 38.4 | 56.4 | 33.7 | 44.9 | 45.8 |
| | LoRA | 48.2 | 66.7 | 45.0 | 60.0 | 37.9 | 46.7 | 55.5 | 40.5 | 55.6 | 35.9 | 51.6 | **46.8** |
| | Adapters | 43.7 | 66.0 | 44.3 | 40.6 | 38.3 | 43.7 | 53.3 | 34.6 | 52.4 | 34.5 | 46.6 | 43.7 |
| | *ours* | 47.4 | 69.1 | 47.6 | 63.1 | 40.0 | 47.7 | 55.3 | 41.0 | 54.5 | 34.7 | **53.4** | **46.6** |

Table 5: **Learning German:** we use the same setting as in Table 4 except that we learn German.

| Model | Method | Forgetting: English | | | | | Learning: German | | | | | Task avg. | |
|---|---|---|---|---|---|---|---|---|---|---|---|---|---|
| | | BeleBele (0) | HellaS (0) | Arc C (5) | CSQA (5) | MMLU (5) | BeleBele (0) | HellaS (0) | Arc C (5) | CSQA (5) | MMLU (5) | English | German |
| Transf-EN | Backbone | 43.0 | 60.3 | 40.5 | 56.5 | 34.5 | 33.8 | 30.2 | 24.0 | 44.3 | 27.8 | **47.0** | 32.0 |
| | finetuning | 40.2 | 44.4 | 32.5 | 44.0 | 30.6 | 42.9 | 44.0 | 34.0 | 69.5 | 30.3 | 38.4 | **44.1** |
| | LoRA | 43.4 | 52.0 | 36.5 | 50.6 | 32.6 | 40.8 | 40.9 | 30.4 | 64.0 | 29.9 | 43.0 | 41.2 |
| | Adapters | 42.6 | 56.6 | 38.4 | 51.4 | 33.5 | 41.6 | 41.2 | 31.7 | 64.2 | 30.7 | 44.5 | **41.9** |
| | *ours* | 42.9 | 59.5 | 40.5 | 56.3 | 34.3 | 39.7 | 41.5 | 31.2 | 62.3 | 30.5 | **46.7** | 41.0 |
| Transf-ML | Backbone | 41.2 | 59.4 | 40.6 | 57.4 | 34.4 | 42.6 | 44.4 | 32.3 | 56.7 | 30.2 | **46.6** | 41.2 |
| | finetuning | 42.1 | 50.7 | 37.8 | 51.5 | 32.6 | 42.3 | 47.5 | 35.6 | 70.2 | 31.8 | 42.9 | **45.5** |
| | LoRA | 42.7 | 56.8 | 38.0 | 56.2 | 34.1 | 41.9 | 45.6 | 35.8 | 65.2 | 31.2 | 45.5 | **43.9** |
| | Adapters | 41.2 | 58.0 | 39.0 | 55.1 | 34.6 | 39.3 | 43.3 | 32.3 | 56.0 | 29.7 | 45.6 | 40.1 |
| | *ours* | 42.1 | 58.9 | 41.6 | 57.7 | 34.5 | 42.8 | 46.3 | 35.7 | 62.6 | 30.6 | **47.0** | 43.6 |
| Gemma-2B | Backbone | 46.3 | 69.7 | 47.0 | 63.0 | 39.9 | 42.4 | 45.6 | 34.4 | 51.9 | 32.3 | **53.2** | 41.3 |
| | finetuning | 45.1 | 55.1 | 40.5 | 54.4 | 34.3 | 44.9 | 50.1 | 37.4 | 68.8 | 33.1 | 45.9 | **46.8** |
| | LoRA | 47.3 | 65.8 | 45.7 | 61.4 | 37.9 | 44.7 | 49.3 | 38.8 | 67.4 | 34.8 | 51.6 | **47.0** |
| | Adapters | 45.9 | 65.9 | 47.4 | 60.9 | 39.5 | 44.7 | 46.9 | 36.5 | 63.6 | 34.5 | 51.9 | 45.2 |
| | *ours* | 47.7 | 67.5 | 49.0 | 64.6 | 39.5 | 45.1 | 49.6 | 37.9 | 64.0 | 34.6 | **53.6** | 46.2 |

Table 6: **Gating, activation and local loss.** For the down-stream tasks, we report the mean of all tasks.

| Gate | Gate losses $\ell_1$ | CE | Our init. | Perplexity EN | FR | Tasks EN | FR |
|---|---|---|---|---|---|---|---|
| ∅ | | | | 0.684 | 0.812 | 45.0 | 40.4 |
| | ✓ | | | 0.681 | 0.813 | 45.5 | 40.6 |
| | ✓ | | ✓ | 0.697 | 0.812 | 45.6 | 41.0 |
| Sigmoid | | ✓ | | 0.668 | 0.810 | 45.7 | 41.1 |
| | | ✓ | ✓ | 0.677 | 0.800 | 45.3 | 41.5 |
| | ✓ | ✓ | ✓ | 0.667 | 0.800 | 45.7 | 41.8 |
| ELU | ✓ | | | 0.670 | 0.813 | 46.1 | 40.0 |
| | | | ✓ | 0.674 | 0.792 | 46.7 | 42.3 |
| | ✓ | | ✓ | 0.668 | 0.793 | 46.5 | 41.2 |
| Backbone baseline | | | | 0.663 | 1.175 | 47.0 | 31.6 |

Table 7: **Ablation:** $\alpha$ governs the strength of the $\ell_1$ local loss. See Table 6 for more comparisons.

| $\alpha$ | Perplexity EN | FR | Tasks avg. EN | FR |
|---|---|---|---|---|
| 0 | 0.674 | 0.792 | 46.7 | 42.3 |
| 1 | 0.674 | 0.792 | 46.4 | 42.0 |
| 10 | 0.671 | 0.792 | 46.4 | 42.1 |
| 100 | 0.668 | 0.793 | 46.5 | 41.2 |
| 1000 | 0.671 | 0.793 | 45.9 | 40.8 |

Table 8: **Trade-offs with $\neq$ learning rates.** This hyperparameter alllows us to favor learning *vs.* forgetting or vice-versa. These results are the ones reported in Figure 1.

| Method | LR | Perplexity EN | FR | Tasks avg. EN | FR |
|---|---|---|---|---|---|
| Backbone | 0 | 0.663 | 1.175 | 47.0 | 31.6 |
| Fine-tuning | $2 \cdot 10^{-5}$ | 0.678 | 0.963 | 45.3 | 36.3 |
| | $5 \cdot 10^{-5}$ | 0.705 | 0.840 | 35.9 | 42.1 |
| | $2 \cdot 10^{-4}$ | 0.811 | 0.758 | 39.0 | 42.8 |
| | $5 \cdot 10^{-4}$ | 0.874 | 0.757 | 35.9 | 42.1 |
| LoRA | $2 \cdot 10^{-5}$ | 0.716 | 0.842 | 44.2 | 40.0 |
| | $5 \cdot 10^{-5}$ | 0.730 | 0.818 | 43.7 | 40.8 |
| | $2 \cdot 10^{-4}$ | 0.745 | 0.802 | 41.8 | 41.4 |
| | $5 \cdot 10^{-4}$ | 0.755 | 0.806 | 41.5 | 42.3 |
| Vanilla adapters | $2 \cdot 10^{-5}$ | 0.686 | 0.830 | 45.0 | 39.5 |
| | $5 \cdot 10^{-5}$ | 0.687 | 0.812 | 45.0 | 40.4 |
| | $2 \cdot 10^{-4}$ | 0.689 | 0.796 | 45.8 | 42.2 |
| | $5 \cdot 10^{-4}$ | 0.688 | 0.795 | 45.4 | 42.2 |
| Neutral residues | $2 \cdot 10^{-5}$ | 0.667 | 0.802 | 46.7 | 41.0 |
| | $5 \cdot 10^{-5}$ | 0.668 | 0.793 | 46.5 | 41.2 |
| | $2 \cdot 10^{-4}$ | 0.670 | 0.789 | 46.8 | 41.2 |
| | $5 \cdot 10^{-4}$ | 0.670 | 0.790 | 46.2 | 41.6 |

## 4.4 ABLATIONS

**Ablation of architectural components and losses.** The low-variance initialization proposed in Section 3 improves the results in the context of gated adapters. Compared to He's initialization, it favors training the gating faster that the other weights. This hypothesis is compatible with the observation that this initialization is detrimental without gating (∅).

**Ablation learning rate.** Table 8 reports the trade-offs achievable by different techniques when varying the finetuning learning rates, see also Figure 1. The results of our method are remarkably stable across a large range of learning rate (one order of magnitude from $5 \cdot 10^{-5}$ and $5 \cdot 10^{-4}$), in sharp contrast with Fine-tuning and LoRA, which are highly sensitive to this parameter.

**Ablation coefficient $\alpha$.** The perplexity results in Table 7 show that this weight gives a trade-off between learning and not forgetting, which is best optimized when selected $\alpha = 100$. We retain this value retained in all our experiments. This choice does not translate to downstream tasks in this case, yet Table 6 shows that using the $\ell_1$ is preferable in other contexts (with no gating or sigmoid).

## 5 CONCLUSION

This paper has explored the feasibility and effectiveness of extending existing foundation models to incorporate new capabilities or knowledge without the need for resource-intensive retraining from scratch. By building upon the concept of adapters, which add new parameters to a pretrained model, we have demonstrated that it is possible to extend a model without compromising its original knowledge, thereby offering a more sustainable approach than retraining from scratch. Our study focused on the use-case of adding a new language to a pretrained model and evaluated the performance of the extended model on both training criteria and downstream tasks. The findings highlight several critical factors that contribute to the successful extension of a model while mitigating the issue of catastrophic forgetting. These factors include the strategic use of mixed training data, an adapter gating mechanism coupled with a local loss, and the importance of near-identity initialization.

## A  ARCHITECTURE DETAILS AND HYPER-PARAMETER SETTINGS

Table 9: Pre-trained models: we consider two vanilla transformers that we have pretrained on two different corpus for which we know the data distribution: English only or MultiLingual with a majority of English. We also considered the Gemma 2B model (Gemma et al., 2024), for which we do not know the distribution. For this model, we refer to their paper for training hyper-parameters.

| | Transformer English | Transformer Multilingual | Gemma 2B |
|---|---|---|---|
| number of layers $L$ | 16 | 16 | 18 |
| working dimensionality | 2048 | 2048 | 2048 |
| number of heads | 128 | 128 | 256 |
| dimension FFN latent space | 5632 | 5632 | 16384 |
| activation type | gated SILU | gated SILU | gated GELU |
| normalization | | RMS pre-normalization | |
| group query | | | ✓ |
| pretraining context length | 4096 | 4096 | 8192 |
| | Hyper-parameters for training the extended model | | |
| batch size | 64 | 64 | 8 |
| context length | 4096 | 4096 | 4096 |
| #training steps | 100000 | 100000 | 150000 |
| AdamW: $\beta_1$ | 0.9 | 0.9 | 0.9 |
| AdamW: $\beta_2$ | 0.95 | 0.95 | 0.95 |

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
