# OpenReview forum: "Neutral residues: revisiting adapters for model extension"
_ICLR.cc/2025/Conference — Submitted to ICLR 2025_

### Official Review · Reviewer_1d3W · 2024-11-01

**Soundness:** 1
**Presentation:** 1
**Contribution:** 3
**Rating:** 5
**Confidence:** 3

**Summary:**

this paper proposes a new adapter architecture that is designed to extend a pretrained model to new domain/language by continue training on new data mixture while freezing the backbone model. The goal is to improve the model performance on new language while incurring minimal forgetting on the pretraining domain/language. The adapter contains several gating mechanism as seen in Figure 2 of the paper. Experiments are done comparing to LoRA, vanilla adapters, full fine-tuning on both open-sourced and closed-sourced models which shows that the proposed method has the best trade-off.

**Strengths:**

1. this paper addresses the problem of efficient adaptation of LLMs to new knowledge without forgetting, which is an important problem for practical usages of these models.
2. the proposed architecture is relatively novel, although the presentation is lacking.
3. The paper includes thorough evaluations of factors like initialization, data mixing, and architecture choices.

**Weaknesses:**

1. this paper is not very well-written so it is difficult to fully asses the content. Section 3 discusses adapter gating and local loss, but I still don't fully understand what each component is like. It is better to write down how the input is transformed through the adapter layer using math formulas.
2. there are some architecture choices that are not clearly explained. Why did you use Silu and Elu activations?
3. on line 315 the authors mentions that the training batch size is 64 and 8, which is quite small. This might make full fine-tuning more unstable. This might not be a fair comparison between different methods.
4. In table 8 the authors show the trade-off between different learning rates, but it's not clear what data mixture it's using. The percentage of new data can affect the conclusion too.

**Questions:**

1. is there ablations about different activation choices?Why did you use Silu and Elu activations?
2. does your method still works best if the amount of training data is less than what's used in the experiments?

---

> ### Author Response · Authors · 2024-12-02
> **Response to Reviewer 1d3W**
>
> First, we would like to thank the reviewer for their feedback on our paper!
>
> **W1. This paper is not very well-written so it is difficult to fully assess the content.**
>
> In the subsection **Adapter gating & local loss**, Adapter Gating refers to the fact that we added a gating to classical adapters in order to distinguish between the pretraining distribution and the learned one as depicted in figure 2. To do so, this gating can be trained using two local losses ($L\_{gating}$):
>
>
> - A classification loss to distinguish the two distributions. The forward formula of the gated adapter in this case is :
>
> $$
> Adapter_\text{gating}(X) = \sigma (\text{proj}(X)) \cdot \left(W\_{out} \left( \text{SiLU}(W\_g X) \odot (W\_{in} X) \right) \right)
> $$
>
>
> - A $\ell_1$ loss applied to the output of the adapters. Actually the $\ell_1$ loss of the outputs is divided by the hidden dimension of the model to get the final loss ($L\_{gating}$). This is done because we wanted something more robust to the hidden dimension of the model. Regarding the activation function of the added gating, our experiments showed us that the activation function leading to the best performance both in learning and forgetting is Elu. We are running additional experiments to support this claim. The forward formula of the gated adapter in this case is:
>
> $$
> Adapter_\text{gating}(X) = \text{ELu} (\text{proj}(X)) \cdot \left(W\_{out} \left( \text{SiLU}(W\_g X) \odot (W\_{in} X) \right) \right)
> $$
>
>
> where :
> - $\sigma $ is the Sigmoid activation function.
> - $ \odot $ is the Element-wise multiplication.
>
> Those local losses are computed for each adapter and meant throughout all transformer blocks to get $L\_{gating}$ which is combined in each of the previous case with the language modelling loss $L\_{LM} $ according to a coefficient 𝛂:
>
> $$
>  L\_{training} = L\_{LM} + \alpha  \cdot  L_\{Gating}.
> $$
>
> We will work on making it clear for a future version of the paper.
>
> **W2. There are some architecture choices that are not clearly explained. Why did you use Silu and Elu activations?**
>
> The SiLU activation is used to implement the Gated linear unit from (N Shazeer · 2020, https://arxiv.org/pdf/2002.05202) in the adapter. We chose this activation for the adapter because it was the one used in the MLP backbone of the Transf-EN and Transf-ML model. For the Gemma model the activation we used is GeLU as it is one used in the backbone. ELu is used for the additional gating previously discussed. An ablation study will be added to support this choice in a revision of the paper.
>
> **W3. On line 315 the authors mentions that the training batch size is 64 and 8, which is quite small.**
>
> - The choice of a batch size (bsz) of 64 with a context length of 4096 is a standard choice. For example this was used in the LLaMA2 paper (https://arxiv.org/pdf/2307.09288, bsz=64, context = 4096)  and in the Direct Preference Optimization paper (https://arxiv.org/pdf/2305.18290 , bsz=64).
> - We agree that bsz=8, used for Gemma, might be too small for the experiments. We are currently running experiments on a batch size of 64 to strengthen our results on this model.
>
> **W4. In table 8 the authors show the trade-off between different learning rates, but it's not clear what data mixture it's using.**
>
> The data mixture used is 10% of English.  Thanks for that remark,  we will clarify it in a future version of the paper.
>
> **Question: Does your method still work best if the amount of training data is less than what's used in the experiments ?**
>
> We have not done experiments on a smaller proportion of English data. We believe that changing the amount of English data would have a similar effect on the learning/forgetting tradeoff of the different methods, and would not change our conclusion.

---

### Official Review · Reviewer_X8Xu · 2024-11-04

**Soundness:** 2
**Presentation:** 2
**Contribution:** 2
**Rating:** 3
**Confidence:** 4

**Summary:**

This paper proposes a new training recipe, with data, architecture, and loss innovations, for adapting a pretrained language model to a new language.

**Strengths:**

The paper addresses an important question: how do we extend a pretrained language model to a new language, without hurting original performance.

**Weaknesses:**

1. More languages are needed to validate the claims. Currently the extensions considered are French and German, which are arguably much more similar to English, syntax- and lexicon-wise, than many other human languages. To show the effectiveness of the proposed method, the authors should consider evaluating on languages that are known to be under-represented (_e.g._, tasks from the XTREME-UP dataset).
2. The assumption of access to a 'similar [pretraining] distribution' (Sec 3) is unrealistic in many cases. However given access to the original checkpoint, there are ways to mitigate forgetting with anchors (e.g., [Agarwal _et al._ (2024)](https://arxiv.org/abs/2306.13649).) The authors should evaluate whether such approaches are effective.

**Questions:**

What are the languages and datasets used to train 'Transformer Multilingual' described in Appendix A?

---

> ### Author Response · Authors · 2024-12-02
> **Response to Reviewer X8Xu**
>
> First, we would like to thank the reviewer for their feedback on our paper!
>
> **W1. More languages are needed to validate the claims**
>
> Thank you for the suggestion, which can help improve the paper. We are currently conducting experiments on languages that differ more significantly from English than French. The results will be included in a future version of the paper.
>
> **W2. The assumption of access to a 'similar [pretraining] distribution' (Sec 3) is unrealistic in many cases.**
>
> We agree with the reviewer that having access to the same distribution as used during pretraining is unrealistic and believe that there was a misunderstanding about what we meant by “similar distribution”. In particular, we do not assume a high level of similarity between the distributions: what we meant is that if the model was pre-trained on English data, a “similar distribution” would be made of English data, as opposed to French data or computer code. Our experimental setup illustrates this difference in training data: for our own model, which was pretrained mostly on data from Common Crawl (and a small amount of data from Wikipedia, StackExchange and scientific articles), we use data from Wikipedia when performing adaptation. We also apply our technique to the Gemma model, for which we do not have any information about its pre-training distribution. Hence, this shows that our method does not require a lot of information about the pre-training data of the model.
> It is unclear to us what is the relation between the problem described in our paper and the one from the paper you mentioned (Agarwal et al., 2024), and how it could be applied to our problem.
>
> **Question: What are the languages and datasets used to train 'Transformer Multilingual' described in Appendix A?**
>
> The languages for  "Transformer Multilingual"  are : English, French, German, Spanish, Italian and Portuguese data. Following previous work such as LLaMA, our pre-training dataset is made of documents from Common Crawl, Wikipedia, StackExchange and scientific articles	from the Semantic Scholar Open Research Corpus.

---

### Official Review · Reviewer_S3Xk · 2024-11-08

**Soundness:** 3
**Presentation:** 3
**Contribution:** 2
**Rating:** 5
**Confidence:** 4

**Summary:**

This paper proposes neutral residues, an improvement on adapters that allows for domain adaptation while preserving the model performance in the original domain. Neutral residues are additional feed-forward gated adapter blocks added to the model, which are optimized such that the if the input is in the pretraining distribution, the adapter output is sparse. The paper studies the effect of factors such as percent of data from the original distribution, adapter architecture, adapter initialization, and adapter training loss.

In experiments for English and multilingual models with French and German finetuning datasets, neutral residues show some improvement over other domain adaptation approaches (full fine-tuning, LoRA, and vanilla adapters) in terms of the trade-off between retaining the model's original knowledge (English perplexity and benchmarks) and learning the new domain (French/German perplexity and benchmarks).

**Strengths:**

- The experiments show some improvement on the trade-off between the original domain and the adaptation domain.
- The idea of optimizing for sparse output if the input follows the training distribution is interesting and seems like a plausible way to maintain the original model performance.

**Weaknesses:**

- Other than the use case provided in the experiments, when is this approach useful instead of something like LoRA or fine-tuning? It seems like the application in the experiments is for a very specific use case where one large domain adaptation would need to be applied, but in real-world settings there are often multiple downstream tasks that would need to be adapted to.
- The additional 20% of parameters seems very high, especially for larger model sizes. It would be valuable to see the results for other domain adaptation methods with varying numbers of additional parameters in Table 3 to provide stronger evidence for the method.
- The experiments would be strengthened by timing comparisons during training and inference. It is not clear to me what the computational cost of this approach is when compared to the other domain adaptation approaches.

**Questions:**

- It would be useful to clarify the introduction and method section to make it more clear what the exact contributions are. Especially in the method section, it is unclear which aspects of the approach are novel.
- A variety of architecture choices were made based on preliminary experiments that are not shown in the paper. It would be useful to include these results in the appendix to support these decisions.

---

> ### Author Response · Authors · 2024-12-02
> **Response to Reviewer S3Xk**
>
> First, we would like to thank the reviewer for their feedback on our paper!
>
> **W1. Other than the use case provided in the experiments, when is this approach useful instead of something like LoRA or fine-tuning?**
>
> In fact, the solution proposed by our paper is for the case of adapting a model to a different new domain. We agree with the fact that models are usually used for specific downstream tasks. For those tasks, approaches such as LoRA might be suitable as they lead to low forgetting and we don’t aim at providing a better solution in that case. But the problem of catastrophic forgetting is more relevant for continual pretraining. It is an important topic as highlighted by **Reviewer X8Xu**  (*"The paper addresses an important question: how do we extend a pretrained language model to a new language, without hurting original performance"*).
>
> **W2. The additional 20% of parameters seems very high, especially for larger model sizes.**
>
> When looking at table 4 and 5 the gap between LoRA and full finetuning is consistent even when using 20% of parameters for the LoRA modules. This is due to the difficulty for the model to learn the new distribution and requires a lot of learnable parameters. We thanks the reviewer for that remark and agree that varying the number of additional parameters for others methods would be valuable.
>
>  We are currently running those experiments for a future version of the paper.
>
> **W3. The experiments would be strengthened by timing comparisons during training and inference.**
>
> - The training time of our method is comparable to that of adapters, as the primary difference lies in the computation of the gating mechanism and its associated loss, which adds negligible overhead compared to training the additional parameters.
>
> - The inference time of our method is nearly identical to that of adapters but slower than LoRA, as our approach incurs computational overhead from additional parameters.
>
>
> Below is the comparison of different methods for Transf-En model with *20% of additional parameters*.
>
> | **Method**      | **Training Time Ratio** | **Inference Time Ratio** |
> |------------------|:-----------------------:|:-------------------------:|
> | LoRA            |          1.00          |           1.00           |
> | Adapters        |          1.05          |           1.14           |
> | Ours            |          1.07          |           1.18           |
> | Fine-tuning     |          1.08          |           1.00           |
>
> **Q1. It would be useful to clarify the introduction and method section to make it more clear what the exact contributions are.**
>
> In the introduction, we thoroughly discussed the contribution of our paper from line 77 to line 90. We emphasised the important factors to reduce forgetting and present the novelty introduced by the paper: the initialisation and the 2 training objectives we study in the paper. We have also specified that the most effective one is *"a sparsity loss whose objective is to ensure that the residual connections output near-zero values when the input follows the pretraining distribution"* .
>
> Thanks for that remark, we will work on making all this clearer for a future version of the paper.
>
> **Q2. A variety of architecture choices were made based on preliminary experiments that are not shown in the paper.**
>
> We will add experiments to justify our architecture choices in a revision of the paper. In particular, we will add results regarding the gating activation function used in the case of sparsity loss.

---

### Meta-Review · Area_Chair_HZ8e · 2024-12-04

**Metareview:**

This paper proposes an improved method of domain adaptation through model extension, which preserves the performance of the model on its original dataset.

The paper has extensive experiments on ideal hyperparameters and data composition, and also does show an improvement in the domain adaptation tradeoff on model forgetting.

The reviewers struggled to understand the paper and suggested improvements in its presentation. They are concerned that the added parameters make this method fairly inefficient. They're also skeptical about the practical use cases for a system designed to adapt to a single domain while maintaining knowledge about a previous domain, and these use cases are not demonstrated sufficiently. They also have asked for additional languages, since only a couple are provided.

**Additional Comments On Reviewer Discussion:**

The authors submitted their rebuttal on  Dec 2, and the reviewers did not have time to discuss it. The authors promised some improvements on the experiments, but do not have the results ready.

---

### Decision · Program_Chairs · 2025-01-22

Reject